# Association of Moral Distress with Anxiety, Depression, and an Intention to Leave among Nurses Working in Intensive Care Units during the COVID-19 Pandemic

**DOI:** 10.3390/healthcare9101377

**Published:** 2021-10-15

**Authors:** Cristina Petrișor, Caius Breazu, Mădălina Doroftei, Ioana Mărieș, Codruța Popescu

**Affiliations:** 1Anesthesia and Intensive Care II Department, “Iuliu Hațieganu” University of Medicine and Pharmacy Cluj-Napoca, 400012 Cluj-Napoca, Romania; ioana.maries@ymail.com; 2Anesthesia and Intensive Care 1 Department, The Clinical Emergency County Hospital Cluj-Napoca, 400006 Cluj-Napoca, Romania; madalinadoroftei5@gmail.com; 3“Prof. Dr. Octavian Fodor” Regional Institute of Gastroenterology and Hepatology, 400162 Cluj-Napoca, Romania; csbreazu@yahoo.com; 4Anesthesia and Intensive Care I Department, “Iuliu Hațieganu” University of Medicine and Pharmacy Cluj-Napoca, 400012 Cluj-Napoca, Romania; 5Department of Practical Abilities, Human Science, “Iuliu Hațieganu” University of Medicine and Pharmacy Cluj-Napoca, 400012 Cluj-Napoca, Romania; codruta_popescu@yahoo.com

**Keywords:** COVID-19, moral distress, ethics, pandemics

## Abstract

Background: Wide geographical variations in depression and anxiety rates related to the ethical climate have been reported during the COVID-19 pandemic in intensive care units (ICUs). The objective was to investigate whether moral distress is associated and has predictive values for depression, anxiety, and intention to resign. Methods: 79 consenting ICU nurses completed MMD-HP and PHQ-4 scales in this cross-sectional study between October 2020–February 2021, after ethical approval. The association between MMD-HP and PHQ-4, and the predictive value of MMD-HP for anxiety, depression, and an intention to leave were analyzed (linear regression and receiver operating characteristics curve analysis). Results: From MMD-HP items, system related factors had highest scores (Kruskal–Wallis test, *p* < 0.0001). MMD-HP and PHQ-4 were weakly correlated (r = 0.41 [0.21–0.58]). MMD-HP and its system-related factors discriminate between nurses with and without depression or anxiety, while system-related factors differentiate those intending to resign (*p* < 0.05). The MMD-HP score had 50 [37.54–62.46] sensitivity with 80.95 [60–92.33] specificity to predict the intention to leave, and 76.12 [64.67–84.73] sensitivity with 58.33 [31.95–80.67] specificity to detect anxiety or depression symptoms. Conclusions: During the COVID-19 pandemic, system-associated factors seem to be the most important root factors inducing moral distress. Moral distress is associated with negative psychological outcomes.

## 1. Introduction

Moral distress is increasingly recognized as an important problem that threatens the integrity of the health care providers [1]. Its intensity might increase with work environment complexity. Moral distress is regarded by ICU nurses as “patient care situations where there is a mismatch or incongruity between expected behaviors of the nurse and his/her personal values or beliefs” [2].

Admission to the ICU is considered one of the most critical events in the life of a patient and family members [3]. Not only do patients and families suffer, but hospital healthcare professionals do too. Caring for patients in critically ill states might also lead to negative outcomes in the staff providing care, when everyday practice overwhelms personal abilities to cope with the complex situations of the ICU environment. Ethical climate is an essential contributor to moral distress [4]. The critical care environment where the sickest and most complex patients are treated put the nurses at the highest risk of moral distress [5]. ICU nurses have higher levels of moral distress than other professionals in other settings facing suffering, death, and ethical dilemmas daily [6,7]. When ethical concepts such as commitment, beneficence, autonomy, justice, nonmaleficence, competency are compromised, moral distress can occur [2]. Nurses have many roles in the care of their patients, also requiring psychological skills. Nurses practicing in the ICU engage in ethical practice; have roles in treating patients and families with dignity; and respect the cultural, religious, and spiritual beliefs while maintaining privacy and confidentiality [8]. These roles of the ICU nurses carry a degree of psychological burden. Ethical awareness or moral sensitivity, as well as the ability to distinguish ethical problems and conflicts, enables nurses to recognize the ethical implications of practice and is important for safe and high-quality nursing care [9,10,11].

Moral distress is linked to direct patient care, personal considerations, and team or hospital workload. Patient-related factors that might lead to moral distress include the care of patients who take end-of-life decisions (with the obligation of prolonging life balanced with the quality of life, decisions to initiate or continue resuscitation), over aggressive and futile treatments, informing and obtaining consent, accepting or rejecting medical procedures, and palliative care (which is not always implemented and other unmet needs of the dying patients) [5,6,12,13]. Team-related factors might be represented by working climate which may include lack of collaboration, conflicts, disrespectful communication, and distrust among team members working in a tense atmosphere [2,6,11,13,14]. The ICU is also a rapidly changing environment with technical skills required and burden of electronic record keeping [15]. Intense mental strain and working conditions under time constraints also characterize the ICU environment, together with possible patient safety events, long working hours, physical overload, and ethical issues. Fluctuations in ICU staff and changes in team members require more resources for training additional staff impacting both patient care and working place predictability [16]. System-related factors are linked to hospitals and healthcare system organizational aspects, together with their direct impact on clinical care. Achieved knowledge that identifies root causes of moral distress is a prerequisite for the management of ICUs, allowing nursing support and education. Moral distress is an ethical root of clinician burnout, impairing wellbeing and impacting job satisfaction and job turnover [4]. Given common determinant factors related to the clinical situations, as well as internal and external constraints, moral distress might be tightly linked to symptoms of depression, anxiety, as well as the intention to leave the job, which are negative personal professional outcomes for staff working in the ICUs. On one hand, the quality of the ethical climate has been linked to an intention to leave and to job turnover [12], while, on the other hand, to frustration, anger, guilt, and/or anxiety in a gradual crescendo of moral residue that could continue for months or years, hampering personal wellbeing [1].

The COVID-19 pandemic, declared in March 2020 by the World Health Organization [17], brought additional psychological burdens for the ICU staff, including fear of being infected, inability to rest, inability to care for family, struggling with difficult emotions, witnessing additional end-of-life decisions more frequently, and the need to make difficult ethical decisions [7,18]. Autonomy, beneficence, non-maleficence, and justice need to be addressed while working in the ICUs, especially during pandemics, as previously described fundamental core ethical principles [19]. During the global COVID-19 pandemic, health care systems have become overwhelmed and psychological pressure upon ICU nurses has increased, a fact highlighted even by nurse suicides [20]. Hospital and healthcare systems have suffered many challenging changes during the COVID-19 pandemic, especially regarding the ICU care of critically ill patients. Rapid measures to combat the healthcare crisis included changes in personal as well as professional lives. COVID-19 experience was difficult, both technically and emotionally [18]. Workload has increased, long-term fatigue has been observed, infection threat and frustration because of high numbers of deaths have been encountered in most countries, while healthcare personnel and their families suffered from separation, as well as social isolation [20,21]. COVID-19 lead to overloading ICUs worldwide with rapidly changing information and overwhelming communication, meaning that nurses had to constantly adapt and do things differently alongside an increased workload and subsequent physical exhaustion [7,16,22,23]. The usual experiences of the ICU nurses might have been increased in the context of the COVID-19 pandemics [22].

Wide geographical variations in depression and anxiety rates related to the ethical climate have already been reported [4]. The differences in the healthcare systems and the pressure upon the hospitals have varied across Europe in different moments of the COVID-19 pandemic. Information regarding moral distress associated with COVID-19 in ICU staff in Romania in 2020 is lacking and the long-term consequences are also not known. Therefore, the aims of our study were to evaluate the levels of moral distress and to appreciate the prevalence of anxiety and depression among ICU nurses during the COVID-19 pandemic in university-affiliated ICUs in Romania; to investigate whether moral distress is associated with depression, anxiety, and an intention to leave the job; to investigate which of the root causes for moral distress (patient-related, team-related, and system-associated factors) are correlated with these negative outcomes; and, finally, to investigate the predictive value of moral distress and years spent working in the ICU for the intention to leave, anxiety, and depression in ICU nurses during the COVID-19 pandemic.

## 2. Materials and Methods

### 2.1. Ethics and Design

The Ethics Committee of the Emergency County Hospital Cluj-Napoca approved this cross-sectional study (No. 19861/13 July 2020) and the surveys were distributed among ICU nurses in academic-affiliated ICUs between October 2020–February 2021, at times when critically ill patients with COVID-19 disease had been cared for in ICUs. After being informed about the purpose of the study, the ICU nurses were invited to complete two scales: The Measure of Moral Distress for Healthcare Professionals (MMD-HP) and The Patient Health Questionnaire for Anxiety and Depression (PHQ-4), as previously described and translated into Romanian [24,25], as well as demographic data including number of years spent working in the ICU and questions about past or present intention to leave their job/current working place. The survey was anonymous and responses were provided on paper.

### 2.2. Instruments

#### 2.2.1. The Measure of Moral Distress for Healthcare Professionals (MMD-HP Score)

The Measure of Moral Distress for Healthcare Professionals (MMD-HP) is a survey instrument that comprises 27 items which reflects three levels of root causes for moral distress linked to ethical issues in current daily practice: patient-level factors, team or unit-level factors, and hospital- or system-associated factors [24]. The items are rated on a Likert scale which evaluates how often that specific factor occurs in practice (frequency: 0 = never, 4 = very frequently) and how distressing that factor is (distress: 0 = none, 4 = very distressing). The product of the two (frequency times distress) gives a composite score per item, with values between 0–16 [24]. The items are clustered in three categories: patient-related factors (items 1,2,3,5,8,10 involving particular patients, family demands, suffering), team- or unit-related factors (items 6,9,11,12,13,14,15,20,21,24,25,26,27, such as poor communication that impacts care and inadequate collaboration between team members), and hospital- or system-related factors (items 4,7,16,17,18,19,22,23, such as chronic poor staffing, administrative pressure, and bed capacity), as described in the methodological paper of Epstein et al. (Table 1) [24].

The total MMD-HP score is obtained from the sum of each item’s score (implicitly the sum of the patient-related, team-related, and system items scores, each representing subscales of the MMD-HP score), with values ranging from 0–432. Higher values for MMD-HP reflect higher levels of moral distress [24]. We have included ICUs that functioned as mixed ICUs in 2020, admitting patients with COVID and non-COVID patients (distinct epidemiological circuits) since April 2020. By October 2020, all of the nurses had been exposed to working with critically ill COVID-19 positive patients. By completing the questionnaire towards the end of 2020 and the beginning of 2021, the nurses’ responses mirror their emotional reactions towards the newly encountered situations.

#### 2.2.2. The Patient Health Questionnaire for Anxiety and Depression (PHQ-4 Score)

The Patient Health Questionnaire for Anxiety and Depression is an ultra-brief screening tool for both anxiety and depression, measured simultaneously because the impact on mental health associated with anxiety is similar to that of depression [25]. The PHQ-4 score is a four-items composite measure that evaluates anxiety using two questions rated 0 (not at all) to 3 (nearly every day) (“Feeling nervous, anxious or on the edge” and “Not being able to stop or control worrying”) and depression using two questions rated similarly (“Feeling down, depressed or hopeless” and “Little interest or pleasure doing things”) [25]. The total score of 3 or more for the first two questions reflects anxiety, while for the last two questions reflects depression, with total scores rated normal (0–2), mild (3–5), moderate (6–8), or severe (9–12) [25]. Nurses with ≥3 points in the first two or last two questions were interpreted as suggesting anxiety or depression, though not definitely diagnostic.

### 2.3. Statistics

The data from all surveys were centralized in an Excel database, and statistical analysis and graphics were performed with GraphPad Prism. As descriptive analysis, for individual items and mean values for each of the three root causes for moral distress data, the mean and standard deviation were calculated and displayed graphically. For statistical analysis, we used non-parametric and parametric tests. Normal distribution was tested with the D’Agostino & Pearson test. For non-normal distribution and less than 30 values analyzed, mean values were compared with the Kruskal–Wallis test. For the comparison of more than 30 values, e.g., comparing MMD-HP total scores or the three subscales among different categories, we used *t*-test or ANOVA.

### 2.4. Outcome Analysis

We investigated the association between the MMD-HP score and PHQ-4 using linear regression analysis and Pearson correlation coefficient. We considered two outcomes to possibly be influenced by moral distress as quantified by using MMD-HP scale: the present intention to leave (binary data Yes or No) and the PHQ-4 score indicative of anxiety and/or depression (scores ≥ 3 for the first two or for the last two questions in the PHQ-4 scale: binary data Yes or No). We evaluated the predictive value of the MMD-HP total score, as well as patient-related, team-related, and system-related subscales, using receiver operating characteristics (ROC) curve analysis (area under the curve (AUC) obtaining *p*-values for AUC, with sensitivity and specificity for individual values on the ROC curves.

## 3. Results

A total number of 83 ICU nurses agreed to complete the survey, but four provided incomplete data, which were excluded. The analysis comprised of the remaining 79 surveys completed by nurses aged 23–64, working in the ICU for 1 to 42 years. We found several types of nursing education programs in our nurses cohort (Table 2).

### 3.1. MMD-HP Analysis

The mean MMD-HP score was 106.63 [95% CI: 93.34–119.93]; standard deviation ± 59.35.

For patient-related factors, the average of the six items’ mean scores was 4.00 [2.16–5.84] ± 1.75. For team-related factors of the thirteen items, the mean score was 3.30 [2.59–4.01] ± 1.17, while for system-related factors, the average of mean scores per item for the eight items was 4.96 [3.29–6.61] ± 1.99 (Figure 1).

The average values of the items’ mean in each of the three categories of moral distress factors have non-normal distribution (D’Agostino & Pearson test, all *p* < 0.0001) and items in the system-related factors group have significantly higher median values compared to patient-related and team-related factors (Kruskal–Wallis test, *p* < 0.0001) (Figure 2).

### 3.2. PHQ-4 Score Analysis

The mean value for PHQ-4 was 2.10 [1.71–2.79] ± 1.73. Anxiety symptoms were present in 11 of the 79 ICU nurses (having ≥ 3 points for the first two questions of the PHQ-4) and depressive symptoms in 2 of the 79 ICU nurses, noting that one of them had both depression and anxiety symptoms. Thus, 12 of the 79 ICU nurses presented depression and/or anxiety symptoms (15.18%).

From the 79 nurses, 58 (73.41%) had a normal PHQ-4 score, 15 (18.98%) had a mild PHQ-4 score, and 6 (7.59%) had a moderate PHQ-4 score. None of them had a severe PHQ-4 score.

For ICU nurses with a normal PHQ-4 score, the mean values for MMD-HP were 91.96 [79.55–104.4] ± 47.2. For those with a mild PHQ-4 score, the mean MMD-HP score was 141.53 [109.5–1736] ± 57.4, while for moderate PHQ-4, the mean MMD-HP was 161 [53.48–268.5] ± 102.46. The mean MMD-HP scores were higher for nurses with a moderate PHQ-4 score compared with mild or normal PHQ-4 scores (ANOVA, F = 8.14, *p* = 0.0006). Post-hoc analysis demonstrated significant differences between mean MMD-HP scores in patients with a normal versus mild PHQ-4 score (*t*-test, *p* = 0.0064), but not between mean MMD-HP for a normal versus moderate PHQ-4 score (*p* = 0.09) nor a mild versus moderate PHQ score (*p* = 0.67).

The association between MMD-HP score and PHQ-4 score is described by a weak positive linear regression relation, with a Pearson coefficient of r = 0.4184 [0.2174 to 0.5854], *p* = 0.0001. The overall regression was statistically significant (R^2^ = 0.17, F (df regression, df residual): F = 16.34, DFn = 1, DFd = 154, *p* < 0.0001). It was found that the MMD-HP score predicted PHQ-4 score.

#### 3.2.1. Predictive Factors for the Current Intention to Leave as Outcome Variable

Intention to leave was predicted using the total MMD-HP score, years spent working in the ICU, and the subscales of the MMD-HP total score as grouped in patient-, team-, and system-related factors. From the 79 nurses, 37 (46.83%) presented past or present intention to leave thoughts. From these, (i) 16 intended to leave in the past but were currently not intending to leave; (ii) 21 nurses considered leaving in the past and were still considering resigning their current job; and (iii) 5 nurses did not consider leaving in the past, but were considering resigning in 2020 when they completed the survey. Thus, a total number of 21 nurses (26.58%) were considering to resign their current position during the COVID-19 pandemic.

ICU nurses with current thoughts to leave presented higher MMD-HP scores, i.e., higher patient-, team-, and system-related scores, but only system-related factors differentiated between nurses intending to leave or not during the COVID-19 pandemic (*p* = 0.042) (Table 3).

When considering current intention to leave as outcome, with a cutoff of 83.50 points on the MMD-HP total score, the sensitivity of MMD-HP was 50 [37.54–62.46] and specificity was 80.95 [60–92.33] to predict the outcome, as highlighted by the ROC analysis (Figure 3).

#### 3.2.2. Predictive Factors for Anxiety and/or Depression Symptoms Identified Using PHQ-4

Prediction of anxiety or depression symptoms was performed using the total MMD-HP score, years spent working in the ICU, and the subscales of the MMD-HP total score as grouped in patient-, team-, and system-related factors. From the 79 nurses, 12 (15.18%) presented ≥3 points in the first or last two questions of the PHQ-4 score. ICU nurses with anxiety or depression symptoms presented significantly higher MMD-HP scores and system-related factors scores (Table 4), but anxiety and depression were not significantly discriminated by patient- or team-related factors, nor by ICU experience. Thus, MMD-HP and its subscale related to system-related factors discriminates between nurses with and without depression or anxiety, suggesting possibly a causative relationship.

With a cutoff of 128.5 points, the sensitivity of the MMD-HP score was 76.12 [64.67–84.73], while specificity was 58.33 [31.95–80.67] to detect anxiety or depression symptoms, as highlighted by the ROC analysis (Figure 4).

## 4. Discussion

The identification of the root causes and level of moral distress in ICU nurses during the COVID-19 pandemic are prerequisites for necessary interventions to ameliorate the ethical climate and reduce negative outcomes. We found that, during the COVID-19 pandemic, moral distress levels, as evaluated using the MMD-HP score in ICU nurses, and especially its system-related factors are associated with depression, anxiety, and an intention to leave the current job, with possible patient and personal consequences. Moral distress, as first defined by Jameton in 1984, is a phenomenon that occurs when nurses cannot carry out what they believe to be ethically appropriate [6,15,26]. Moral distress comes from the inability to act as considered ethically appropriate [6]. This represents an important aspect of working in a complex and challenging environment, such as the ICU where differences in opinions, attitudes, and values can lead to controversies and even conflict. Both the COVID-19 pandemic and the critical care environment are circumstances that generate high psychological risks for healthcare providers [18]. From the MMD-HP score items, we identified system-related factors as having higher mean scores compared to patient- or team-related ones, highlighting the importance of these factors for the generation of moral distress in ICU nurses during the COVID-19 pandemic. Thus, from the root causes of moral distress, system-related factors pose a greater challenge in our ICU nurses and can lead to moral distress during the pandemic. This fact might be directly linked to the additional burden the COVID-19 pandemic brought in the ICU organizational and ethical issues. The mean MMD-HP score was 106.62. This is similar to that reported by Epstein et al. [24] which might mean that moral distress severity might be close before and during the pandemic, possibly because ICU nurses face critical patients in their daily practice, irrespective of pandemics. Still, the root causes may vary in time in such new situations. By exactly pointing out the main causes of moral distress, interventional programs targeting those root causes could be designed. Our ICU staff would possibly have benefited from interventions targeting the organizational aspects of workflow.

During the COVID-19 pandemic, in 2020, working together with nurses from other wards due to staff shortages was decided for our ICU. The rapidly changing structure of the ICUs needed to accommodate higher than usual numbers of patients with high mortality rates, together with the need to implement triage, as well as scarce resources. New protocols needed to be implemented in a short period of time. Employing freshly graduated nurses or mixing with nurses from other wards with no experience carrying for critically ill patients has reduced the level of the skill mix [20,27]. All these have direct consequences on staff wellbeing.

Moral distress is associated with anxiety, depression, and an intention to leave in ICU nurses. Moral distress encountered in the ICU healthcare workers, and especially nurses, brings negative personal, professional, and institutional consequences. Depression, burnout, posttraumatic stress disorder, substance abuse, and even suicidality have been observed [16]. The professional life is negatively impacted by the possible inability to cope with threatening events, impairment of decision-making capabilities, losing capacity for caring, avoiding patient or family contact, which all reduce the quality of care and team communication [28]. Not only do the patients and the healthcare provider suffer, but also the healthcare systems too. The ICU team, the hospital, and the healthcare system face increased fluctuation rates due to the intention to leave resulting from job dissatisfaction. For hospital managers, it became increasingly challenging to retain clinicians in the ICUs and to reduce job turnover [12].

In ICU personnel, the rates of depression were cited as being 8–43.3%, anxiety rates were 16–46.5% [4,7,15], and even higher [22,23], while the intention to leave was approximately 20–40% before the pandemic [1,12]. During the coronavirus (COVID-19) pandemic, rates of depression, anxiety, and peritraumatic dissociation in health care professionals working in the ICU was reported to be high (50.4%, 30.4%, and 32%, respectively), demonstrating a significant burden on mental health [18]. The long-term consequences are not yet known, but deserve the attention of specialists. The chronic impact of the current healthcare crisis on the psychological wellbeing of the medial personnel remains to be established [7].

We designed the present study to better understand the causes of work-related stress in ICU nurses in Romania during the COVID-19 pandemic. The prevalence of symptoms of anxiety, depression, and burnout significantly varied across regions in Europe. Among the determinants of mental health outcomes in ICU staff, the ethical climate in the ICU has been identified as an important causal factor [4]. Psychological burden might present an even wider variation across regions during the COVID-19 pandemic due to differences in organization of the local hospitals and especially the ICUs, while national supply of protective equipment was scarce and national regulations, such as lockdown periods, have also been variable.

In a study by Colleville et al., a moral distress scale was simultaneously evaluated together with an anxiety/depression scale which highlighted the prevalence of anxiety (16%), depression (8%), and an intention to leave the job (16%) in the ICU before the pandemic [15]. We found similar depression and anxiety symptoms rates in 15.18% of our cohort of ICU nurses, possibly because nurses face critical patients daily, irrespective of pandemics.

The moral distress score and its included system-related factors differentiate between ICU nurses which demonstrate depression and/or anxiety symptoms, while system-related factors differentiate nurses who intend to leave current job from those who do not, which suggests that ethical problems associated with moral distress are associated with negative staff outcomes during the COVID-19 pandemic. The total MMD-HP score and the system-related factors demonstrate predictive values for the intention to leave the job and for the occurrence of depression and anxiety, which suggests that, in our cohort of ICU nurses, organizational issues might be more important for these negative personal outcomes compared to the core working activity, i.e., the patient care within an ICU team during the COVID-19 pandemic. Organizational-level interventions that improve work control and emphasize communication and values might also improve job satisfaction and reduce stress, these being crucial roles now, during, and after the COVID-19 pandemic [18].

### Future Directions

Management strategies to improve psychological outcomes of the personnel working in the ICU should include attempts to reduce moral distress. Improvement strategies to reduce moral distress by workplace measures might have a positive impact on patient care and personal psychological wellbeing of the ICU staff, and might improve job satisfaction and retention [5,18]. Past efforts to reduce burnout and job resignation have mainly focused on improving resilience skills [12]. Interventions that have already been described try to build resilience, create a culture of moral resiliency, and improve wellbeing [2,22]. Unfortunately, the numbers of interventional programs, such as debriefings, open communications, dialogue in reflective practices, and organizational support structures, is relatively small and includes limited numbers of nurse participants; however, these programs do suggest possible improvements [2,6].

Are moral distress and its consequences inevitable? Possibly, yes. This is due to the high complexity of the ICU environment. Still, interventional studies are required in order to reduce the level of distress. Factors associated with psychological burden that may be amenable to change by applying strategies to preserve mental wellbeing are of utmost importance for the management of the ICU environment. Knowledge of the root causes for moral distress, as well as the degree and intensity of moral distress and its associations, are important when designing interventions to ameliorate the ethical climate and to increase moral sensitivity and ethical awareness.

The limitations of our study include the modest number of nurses that completed the questionnaires. This might represent a possible selection bias, with data from our region not being eligible for generalizable conclusions. Still, in the context of wide geographical variations due to different healthcare systems and local hospitals organizations, it is important to evaluate these differences when designing interventional studies to reduce moral distress and negative psychological outcomes in ICU healthcare workers. Our cross-sectional study is a descriptive one which highlights associations between ethical issues investigated with MMD-HP and depression or anxiety symptoms, together with the possibilities of predictive values. However, causality is more complex and requires the conduction of different types of studies with larger sample sizes. Our study does not strongly demonstrate a link of causality between ethical issues and depression or anxiety. Other limitations are given by the fact that long-term consequences are not yet available and causal relationships have not been investigated. Furthermore, in this cross-sectional study, we had no comparison to periods before the COVID pandemic among the respondents. This could have highlighted the before and after differences in moral distress severity, as well as rates of anxiety, depression, and intentions to leave, allowing for a better understanding of the causes which need to be addressed.

## 5. Conclusions

During the COVID-19 pandemic in Romania, system-associated factors are the items with highest scores among those included in the MMD-HP moral distress scale as reported by ICU nurses, highlighting the extra ethical issues brought by the new and rapidly changing circumstances which the healthcare system, and specifically ICUs, have faced.

Higher levels of moral distress are positively associated with higher scores on the depression and anxiety scale. Moral distress score and its included system-related factors differentiate between ICU nurses that demonstrate depression and/or anxiety symptoms, while system-related factors differentiate from those intending to leave current job from those who do not. These suggest that ethical problems induced by moral distress might be associated with negative psychological outcomes during the COVID-19 pandemic.

## Figures and Tables

**Figure 1 healthcare-09-01377-f001:**
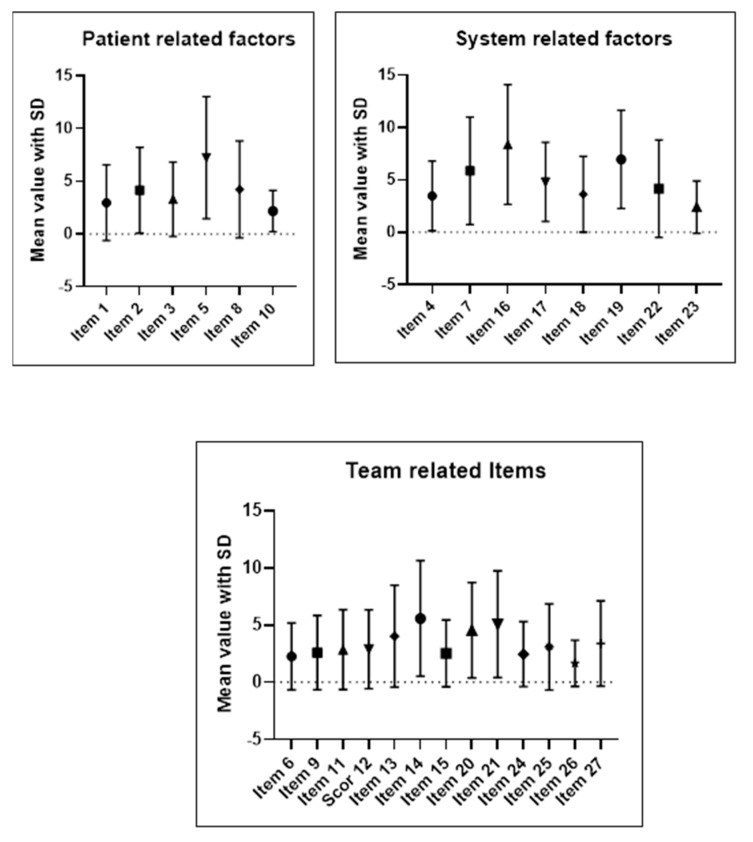
Individual analysis of each item of the MMD-HP score, as grouped in patient-related, team-related, and system-related factors. Results are displayed graphically using means and standard deviation.

**Figure 2 healthcare-09-01377-f002:**
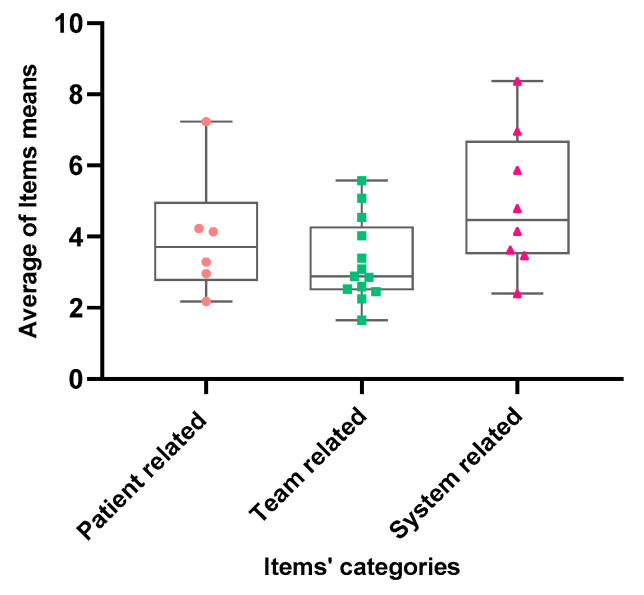
Box plot for distribution of average values for items’ means in patient-related, team-related, and system-related causes of moral distress in our cohort of ICU nurses.

**Figure 3 healthcare-09-01377-f003:**
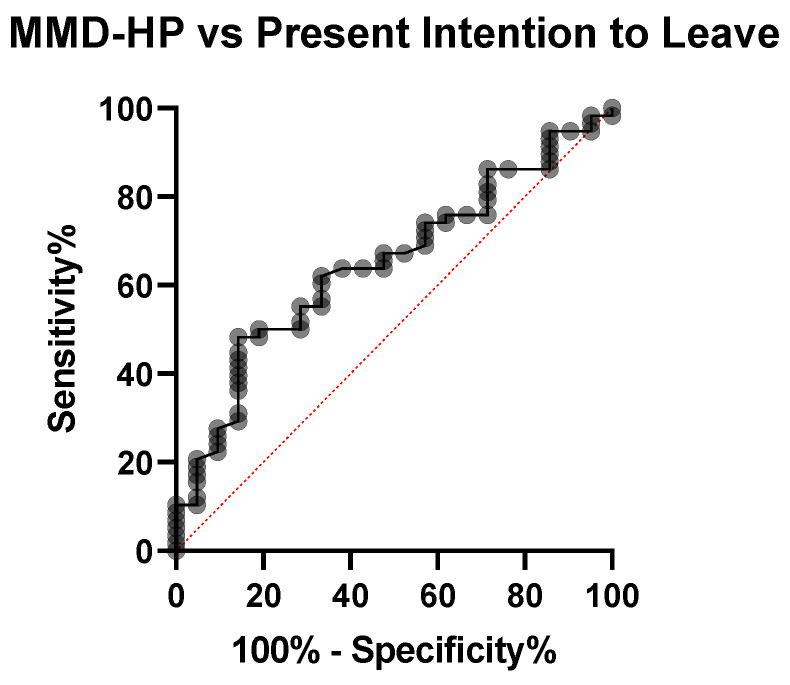
ROC curve of MMD-HP score versus present an intention to leave the current workplace in the ICU.

**Figure 4 healthcare-09-01377-f004:**
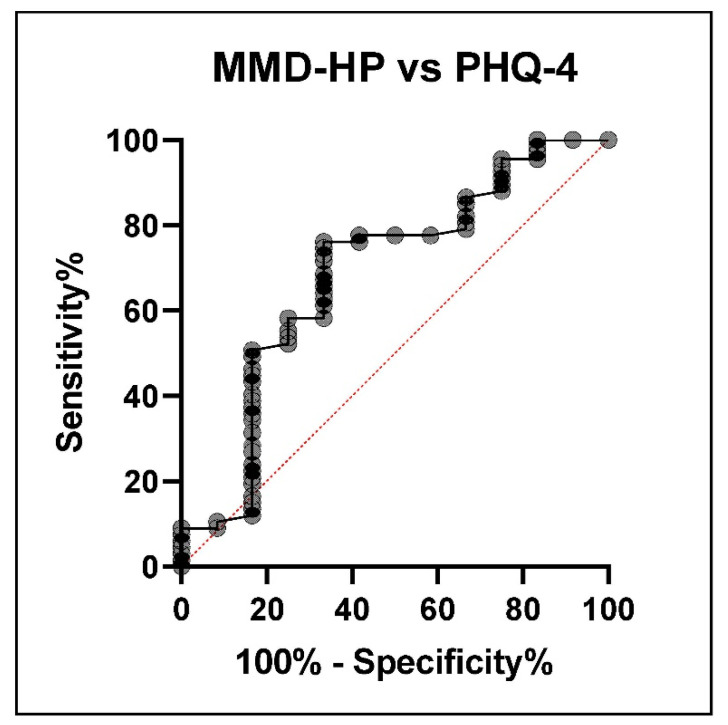
ROC curve of the MMD-HP score versus depression and/or anxiety symptoms in ICU nurses.

**Table 1 healthcare-09-01377-t001:** MMD-HP score items grouped in three root causes for moral distress: patient-, team-, and system- related factors, as described by Epstein et al. [24].

Patient-related factors	1. Witness healthcare providers giving “false hope” to a patient or family.
	2. Follow the family’s insistence to continue aggressive treatment even though I believe it is not in the best interest of the patient.
	3. Feel pressured to order or carry out orders for what I consider to be unnecessary or inappropriate tests and treatments.
	5. Continue to provide aggressive treatment for a person who is most likely to die regardless of this treatment when no one will make a decision to withdraw it.
	8. Participate in care that causes unnecessary suffering or does not adequately relieve pain or symptoms.
	10. Follow a physician’s or family member’s request not to discuss the patient’s prognosis with the patient/family
Team-related factors	6. Be pressured to avoid taking action when I learn that a physician, nurse, or other team colleague has made a medical error and does not report it.
	9. Watch patient care suffer because of a lack of provider continuity.
	11. Witness a violation of a standard of practice or a code of ethics and not feel sufficiently supported to report the violation.
	12. Participate in care that I do not agree with, but do so because of fears of litigation.
	13. Be required to work with other health care team members who are not as competent as patient care requires.
	14. Witness low quality of patient care due to poor team communication.
	15. Feel pressured to ignore situations in which patients have not been given adequate information to ensure informed consent.
	20. Fear retribution if I speak up.
	21. Feel unsafe/bullied among my own colleagues.
	24. Be required to care for patients who have unclear or inconsistent treatment plans or who lack goals of care.
	25. Work within power hierarchies in teams, units, and my institution that compromise patient care.
	26. Participate on a team that gives inconsistent messages to a patient/family.
	27. Work with team members who do not treat vulnerable or stigmatized patients with dignity and respect.
System-related factors	4. Be unable to provide optimal care due to pressures from administrators or insurers to reduce costs.
	7. Be required to care for patients whom I do not feel qualified to care for.
	16. Be required to care for more patients than I can safely care for.
	17. Experience compromised patient care due to lack of resources/equipment/bed capacity.
	18. Experience lack of administrative action or support for a problem that is compromising patient care.
	19. Have excessive documentation requirements that compromise patient care.
	22. Be required to work with abusive patients/family members who are compromising quality of care.
	23. Feel required to overemphasize tasks and productivity or quality measures at the expense of patient care.

**Table 2 healthcare-09-01377-t002:** Characteristics of the population.

Gender (N, %)	Male	8 (10.12%)
	Female	71 (89.87%)
Age (Years) (mean, ±SD)		37.05 (±8.77)
Years of working in ICU (mean, ±SD)		11.12 (±8.49)
Relationship status (N, %)	Single	11 (13.92%)
	In a relationship	14 (17.72%)
	Married	52 (65.82%)
	Divorced	2 (2.53%)
Children (N, %)	Yes	40 (50.63%)
	No	39 (49.36%)
Education (N, %)	Private vocational nursing school (high school + 2 years)	41 (51.89%)
	Faculty of Nursing (high school + 4 years)	36 (45.56%)
	Specializes Master (high school + 4 years faculty + 2 years Master)	2 (2.53%)

**Table 3 healthcare-09-01377-t003:** Mean values for MMD-HP score and subscales, as well as experience of ICU work in nurses presenting a current intention to leave or not.

Outcome	Intention to Leave		*t*-Test	ROC Analysis
Number	Yes (n = 21)	No (n = 58)	*p*-value	AUC [CI 95%]
MMD-HP total score	123 ± 53.84	100.68 ± 60.57	0.12	0.65 [0.52–0.78], *p* = 0.045
Sum of patient related factors items	27.85 ± 15.91	22.63 ± 15.53	0.20	0.61 [0.47–0.74], *p* = 0.13
Sum of team-related factors items	47.42 ± 27.73	41.32 ± 31.25	0.40	0.59 [0.46–0.73], *p* = 0.18
Sum of system-related factors items	47.71 ± 20.47	36.74 ± 20.41	0.042	0.65 [0.52–0.78], *p* = 0.042
Years in ICU	9.23 ± 6.2	9.48 ± 8.81	0.89	0.54 [0.39–0.68], *p* = 0.60

**Table 4 healthcare-09-01377-t004:** Mean values for MMD-HP score and subscales, as well as experience of ICU work in nurses presenting with depression or anxiety symptoms on the PHQ-4 score.

Outcome	Depression or Anxiety PHQ-4		*t*-Test	ROC Analysis
Number	Yes (n = 12)	No (n = 67)	*p*-value	AUC [CI 95%]
MMD-HP total score	144.16 ± 81.17	99.91 ± 52.55	0.016	0.69 [0.51–0.86], *p* = 0.041
Sum of patient related factors items	30.83 ± 20.51	22.80 ± 14.55	0.10	0.60 [0.41–0.80], *p* = 0.23
Sum of team-related factors items	58.66 ± 45.30	40.13 ± 26.27	0.0505	0.61 [0.43–0.80], *p* = 0.19
Sum of system-related factors items	54.66 ± 26.22	36.97 ± 18.76	0.0061	0.70 [0.52–0.88], *p* = 0.023
Years in ICU	10.1 ± 9.24	9.3 ± 7.94	0.77	0.50 [0.29–0.73], *p* = 0.95

## Data Availability

Data available on request.

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
