# Peer review of "Association of Moral Distress with Anxiety, Depression, and an Intention to Leave among Nurses Working in Intensive Care Units during the COVID-19 Pandemic"

_healthcare, 2021, doi:10.3390/healthcare9101377_

Round 1
Reviewer 1 Report
Thank you for the opportunity of reviewing this interesting manuscript. The study describes the impact of moral distress experienced by ICU nurses on symptoms of anxiety, depression and intention to leave. The authors report associations with some aspects of moral distress and the indicated outcomes. While the study of this topic is important to determine appropriate intervention options for healthcare personnel during the COVID-19 pandemic, I feel that the manuscript needs to be improved in several areas as detailed below. A revision by a native English speaker might also contribute to a better overall comprehension of its contents.
Introduction
- The introduction lacked a clear overview of the topic of moral distress in general, its relation with ethical climate, or why it is particularly relevant for their target group of ICU nurses during the COVID-19 pandemic. However, a large part of this information appears in the first part of the discussion (lines 232-311) and should be moved forward.
-Lines 59-60: Wide geographical variations in depression and anxiety rates related to the ethical climate have been reported: This statement could have been elaborated on, especially why this is pertinent to the specific aims for this study (e.g. not yet studied in Romania?)
-Research aim: the presented research aims for this study are not clear. For instance “…to investigate which of the root causes for moral distress (patient related, team related and system associated factors) are correlated with these negative outcomes;” What is the rationale (why is it necessary/interesting) to decompose the concept of moral distress in relation to these outcomes? In addition, what is the difference between the first vs the last study aim?
Methods
-Was the questionnaire presented in a way that made it clear it was specifically aimed at the work situation during the COVID-19 pandemic?
-Reformat table 1 so that the separate items are more “readable”, it would also be useful to use the item numbering that is used on page 2, lines 88-91.
-The description of the outcome “intention to leave” is very brief. How was this question asked to the respondents, to which timeframe does it refer (before/during COVID-19)? Also, why were the scores of past and present intention combined?
-What is meant by Page 3 lines 119-122 (more/less than 30values): Does this refer to mean scores or participants?
-Lines 124-126: The association between MMD-HP score and number of years spent working in the ICU (continuous and PHQ-4 was analyzed using linear regression analysis and Pearson correlation coefficient was calculated. I am not really sure what the authors want to say here?
-Were the associations tested in the linear regression analyses corrected for potential covariates?
Results
-I miss a description of the recruited sample, age, sex, work experience, perhaps also some characteristics in relation to their work situation/hospital.
-Extended reporting on that the average values have a non-normal distribution has no added, the information can be used to choose the appropriate analyses
- It is preferable to speak of “symptoms of depression and/or anxiety” instead of trait, which suggest that it is a stable characteristic of the person
-Lines 170-171: report the results of the anova according to conventional format: 1)the overall F-value of the ANOVA and the corresponding p-value ; 2) the results of the post-hoc comparisons (if the p-value was statistically significant).
- Figure 3 does not add much to the presented information
-Concerning the regression plot for MMD-HP and PHQ-4 likewise the figure does not add much beyond the text. Give a p-value for the Pearson coefficient
-lines187-189 “Thus, MMD-HP and subscales related to system and patient related factors discriminate between nurses with and without intention to leave, suggesting possibly causative relationship”.
-Beyond my previous stated concerns on the outcome ‘intention to leave’, the associations are based on cross-sectional analyses that have not been adjusted for confounders. To interpret this as a possible causal relation is a big over interpretation of the evidence at hand. Same later on when the associations between phq4- scores predicting MMD-HP scores are discussed
-Why was the combination of high score on anxiety + depression used as a binary criterion for the ROC scores and not the moderate PHQ44 overall score?
Discussion
-Overall the discussion section is difficult to follow as it is unorganized and has several repetitions
-The moral distress of the ICU nurses in this study did not seem overly high. Also, comparing the mean score to another study done before the pandemic in another country does not give a good comparison of how the moral distress of your study population has evolved. Same for the conclusion about the “root causes”: as we do not know which factors contributed most before COVID-19, the conclusion that they vary in time is not supported by the data.
-The suggestion that improving the ethical climate through intervention is valid, but it lacks a link to the specific outcomes of this study. If you know that institutional factors contributed most, what kind of interventions would you suggest?
Author Response
Please see the attachment with the point-by-point responses.

Reviewer 2 Report
Dear authors and editor,
I consider that the research is correct, clear and precise.
The abstract is clear and concise, I find it very interesting to add the statistical data because it allows the reader to easily understand the typology and nature of the data. The title and the keywords are coherent with the topic. It is true that the title is not flashy, it is not a slogan but it summarizes perfectly the nature of the topic. In addition, this type of title helps researchers to find information. I congratulate the authors for opting for a simple, clear and honest title, and not giving in to fashions of using titles with a "hook".
As for the introductory section. It is correct, simple and easy to understand, but more bibliographical references are missing. Those that exist are not sufficient. I understand that the authors wanted to simplify and make the reading easier, but the review needs to be supported by more references. The authors make a typical but correctable error. They use bibliographic references in the discussion that do not appear in the introduction. That is, they should expand the introduction, add all the references of the discussion and also look for about 10 new references (last 5 years) so that the introduction is well updated.
Regarding materials and methods. I congratulate the authors for putting the ethical code and explaining it. The methodology denotes transparency, each of the procedures has been detailed. There are only two details to be solved. First, referencing the questionnaires (add it as another reference). Second, the adaptation of the test to the cultural context in which the study was carried out must be made explicit. This test will appear between " " and in the mother tongue. Another important element. I understand that as health care professionals we are used to knowing the tests by their acronyms, but in research we must structure this information. Under "2. Materials and Methods" line 70, add a new line "Ethics and design" to head the text from line 70 to 78. Before "MMD-HP score" add a subsection that says "instruments" and heads the description of all instruments. When presenting the instrument, put the full name first (with its reference and the version for the culture of your region) and then the acronym of the instrument in parentheses. In addition, add more information about the sample, the type of procedure, mean age of participants, gender or sex, and any sociodemographic information that helps to understand the context. Finally, to facilitate understanding of the design and procedure, create a summary figure showing the dates, instruments, sample, etc. In this way, the reader will understand the research at a glance.
As for the results. These are simple and coherent. Add in Figure 4, the significance of the regression. If you cannot modify the image, just add it in line 176, after "with Pearson coefficient r = 0.4184 [0.2174 to 0.5854] (Figure 4)", it is enough.
As for the discussions. These should be revised with the changes in the introduction. Once you have those changes made, update and supplement the discussions. The current development is correct, but to bring out the full potential of the research you will have to make small updates with the new bibliographic references. The "Future directions" section is very carefully done, I congratulate you, would it be possible to develop in this section direct actions on improvement strategies? Perhaps add a couple of concise and tight examples from other researchers (if any) that would encourage other researchers to bet on these issues.
Regarding conclusions. Personally, I think the conclusions should be concise and clear as they are. But it is possible that other reviewers will tell them to lengthen them. In any case, although I really like lines 450 and 451 "Higher levels of moral distress are positively associated with higher scores on the de- 450 pressure and anxiety scale" they are a bit sparse, could you give them a bit more context or connectors with the paragraphs above and below?
Finally, I want to congratulate you on your rigor, transparency and simplicity. Conducting research with people is very complex, but in health care settings it is even more so, as the internal structure of the health care system complicates research. I encourage you to always maintain the ethical protocols, as you have done, and to be committed to transparency in results and methods. Although the research career is very hard, research like yours helps the rest of us to maintain an attitude of professional ethics.
Your manuscript is accepted by me with minor revisions.
Thank you for your efforts
Author Response
Please see the attachment with the point-by-point responses. Thank you.

Reviewer 3 Report
I have completed my review of the manuscript. This topic is a very interesting and very important segment for nursing.
On the beginning, I want to give my compliments to the authors for a interesting study. Introduction and method are well written. Despite that, this study has some limitations in results and discussion.
- Results: I think that it is necessary to give the descriptive statistic of nurses (age, gender, length of services etc.)
- The discussion part needs revision. It is very long and somehow difficult to follow despite subtitle. I miss the "red line". It would be improved if you start to describe the main results of your study. Thought all discussion the authors describe the theoretical and scientific concept of the own study topic, but do not relate it to their results. The discussion should clearly explain WHY the results are just like that and what they mean. I think that it is necessary to better connect your results with results from other studies (which are described in discussion) and explain why. Subtitles can facilitate the above, but if the discussion is fluently written subtitles will not be needed.
- In limitation to add the limitation type of study (for example cross sectional- not detects causality in results), control groups other nurses out of ICU during pandemic.
Author Response
Please see the attachment with the responses.

Round 2
Reviewer 1 Report
Thank you for this revised manuscript. Many of the issues I highlighted for the previous version have been addressed. However, I have some remaining comments that need to be taken into account before it is fit for publication
introduction : The content of the introduction has been greatly improved now that information from the discussion section has been moved forward. However, it would benefit from an additional proof reading, as some paragraph still do not flow nicely due to unnecessary repetition (e.g. between lines 44 and 54, and again lines 70-72)
It is not clear at which point the system related aspects of moral distress are discussed.
Were the prevalence of anxiety and depression among Romanian IC nurses already reported in previous studies (lines 128-129)? It is still not clear if this information is also lacking
Methods
I still have some issues with the combined "intention to leave" measure. The aim of this study is to see if moral distress in times of COVID is associated with the intention to leave. When combining the questions in 1 measure, it is no longer possible to ascertain this effect.
I do not understand the rationale "We have combined this intention from the past or the present because it might be a link to the personality/ psychological traits of the nurses." I would run at least a sensitivity analysis where the 2 questions are split and associations are tested for each of them with your intended outcomes.
Results
The results of the association between MMD-HP score and PHQ-4 scoreare reported as a pearson correlation coefficient. If linear regression has been run, reporting should be more complete for instance:
The overall regression was statistically significant (R2 = [R2 value], F(df regression, df residual) = [F-value], p = [p-value]). It was found that [predictor variable] significantly predicted [response variable] (β = [β-value], p = [p-value]).
I understand that the ROC curves are based on binary scores, but it would have been possible to create a dichotmous variable (PHQ-4 >=moderate score yes vs no)
Discussion
it might be good to start your discussion with a succint repetition of your study aims and the main results.
Lines 360-361 are a repetition of the onces above
Lines 414-415 seem to be unfinished?
Lines 483: do you mean to say similar prevalence rates?
Author Response
Please see the attachement. Thank you very much.

This manuscript is a resubmission of an earlier submission. The following is a list of the peer review reports and author responses from that submission.